# Extraction of driving behavior primitives considering driver expectation and vehicle dynamics

Yuanyuan Ren[1]☯, Xiaotong Cui[2]☯, Xuelian Zheng📧[1]☯*, Xiansheng Li[1]‡, Jianfeng Xi[1]‡

1 Transportation College, Jilin University, Changchun, Jilin, China, 2 School of Automotive Engineering, Changchun Institute of Technology, Changchun, Jilin, China

☯ These authors contributed equally to this work.
‡ XL and JX also contributed equally to this work.
* zhengxuelian@jlu.edu.cn

## Abstract

Extracting meaningful driving behavior primitives is crucial for fine-grained behavior analysis. To develop primitives that more closely mirror the hierarchical nature of human driving, this paper introduces a novel framework and carries out on a 16 drivers' dataset. The proposed method operates in two synergistic stages. First, for behavior segmentation, a multi-type feature space is constructed to capture both the objective vehicle motion states and the subjective driver expectations on vehicle performance. This space is then input into a Bayesian Model-based Agglomerative Sequence Segmentation (BMASS) model to achieve precise segmentation. Segment durations and the positioning of segmentation points serve as key metrics to assess the quality of behavior segmentation. Next, for primitive clustering, a Variable Coupling-based Latent Dirichlet Allocation (VC-LDA) method is proposed. The core innovation of VC-LDA lies in a features-coupling-aware discretization process. By considering the non-linear coupling and temporal asynchrony among features in the multi-type space, this process yields driving states with enhanced physical interpretability, providing a high-quality foundation for LDA clustering. Experimental results demonstrate that the VC-LDA model significantly outperforms the GMM-LDA, achieving a much lower perplexity and exhibiting higher intra-class compactness in topic-word distributions. This framework offers an automated and efficient pathway to understand and model driver behavioral patterns, providing valuable insights for the development of ADAS and AVs.

## 1. Introduction

Driving behavior refers to the decisions and reactions taken by drivers in response to traffic environments, reflecting their distinct behavioral patterns. Analysis of driving behavior includes topics such as driving style [1–3], behavior modeling [4,5], and

**Data availability statement:** All relevant data are within the paper and its Supporting information files.

**Funding:** This research was funded by the National Key R&D Program of China (2023YFC3009600).

**Competing interests:** The authors have declared that no competing interests exist.

multi-vehicle interaction trajectory planning [6]. Understanding these behavioral characteristics is crucial for gaining insights into driving habits, improving traffic safety and promoting the development of autonomous driving technologies [6–8].

Traditional driving behavior studies most rely on feature construction to extract information within the data, achieving a macroscopic description [9,10]. However, drivers typically achieve their goals through a combination of simple actions, which reflect the decision-making strategies employed when facing complex situations [11,12]. Therefore, to gain deeper insights into this decision-making process, it is essential to segment high-dimensional, long-duration time series driving-behavior into low-dimensional, low-frequency, and easily interpretable segments, referred to as driving behavior primitives [13,14].

Three main approaches are typically used for behavior segmentation: rule-based, template-based, and data-driven methods. Rule-based segmentation relies on empirical thresholds and tends to be somewhat subjective. Template-based methods determine segment boundaries by comparing data to predefined templates, which can be computationally expensive when handling high-dimensional data [15]. In contrast, data-driven methods segment sequence by analyzing its intrinsic structure, which is more efficient and can get the segmentation results that align more closely with human cognition [16]. For example, maximize the difference between segments or minimize the difference within segments based on the sliding window method to determine the segments boundaries [5,17]. Additionally, Agamennoni *et al*. proposed a Bayesian model-based agglomerative sequence segmentation (BMASS) algorithm, which characterizes data through likelihood [11,18]. This method demonstrates significant advantages in both accuracy and computational efficiency. Many scholars have made further research based on this approach and yielded good performance [7,8,19].

The segmentation often results in many segments, making it difficult to interpret. Clustering is required to facilitate the intuitive recognition of behavior primitives from the resulting segments [20]. Both hard and soft clustering approaches are utilized until now. Hard clustering methods usually solve the problem of inconsistent durations of segments by discretizing the time series into static statistical features, such as extracting static features through auto-regression and moving average model (ARAM), followed by the application of *k*-means clustering [21,22]. Several scholars have improved these algorithms to better handle segments with different durations, among which *k*-means clustering based on dynamic time warping (DTW) and *k*-shape clustering based on shape-based distance (SBD) have achieved better performance [12,23,24]. Compared to hard clustering methods, soft clustering offers more flexibility in handling behavior segments. The soft clustering methods based on Latent Dirichlet Allocation (LDA) can effectively capture the underlying structure of segments by performing segments clustering according to data distribution. Extended LDA models, such as GMM-LDA, GW-LDA, and mLDA, were developed to handle the discrete inputs required by LDA, as behavior segment is continuous time-series [11,14,25–27]. Many scholars have made further research on this basis and

confirmed that the LDA model exhibits strong adaptability and expressive power, making it an increasingly popular choice in driving behavior segments cluster.

Based on the summary of current research, the following key issues in driving behavior primitive inference remain to be addressed:

(1) In multi-dimensional driving behavior segmentation by BMASS, most existing approaches use raw vehicle motion variables as segmentation features, while ignoring the implicit driver intent or expectation, which is the actual internal factor governing driving maneuvers. This omission is likely to result in inaccurate and semantically misaligned segmentation outcomes.

(2) During segment clustering using extensions of Latent Dirichlet Allocation (LDA), continuous driving behavior segments must first be discretized. However, current discretization methods only consider the numerical distribution characteristics of driving variables and overlook the inherent coupling relationships among them. This leads to an incomplete discrete representation of driving behavior, which in turn undermines the quality of subsequent LDA-based clustering.

To accurately infer driving behavior primitives, this paper proposes a two-stage unsupervised framework that integrates BMASS and VC-LDA model. The main contributions are summarized as follows:

(1) In the driving behavior segmentation stage, a multi-type feature space is constructed by combining basic vehicle motion features with constructed features reflecting driver intent or expectations on driving performance. This feature space is then used as input to BMASS for segmentation. Experimental results verify that the inclusion of intent-aware features allows for more accurate and behaviorally meaningful segment identification.

(2) A VC-LDA model is developed to cluster behavior segments. The model first analyzes coupling relationships between basic and constructed features, and discretizes continuous driving behavior accordingly. Based on this discretization, standard LDA is then applied for cluster inference. The proposed VC-LDA effectively mitigates issues such as over-concentrated or fragmented segment definitions, leading to substantially improved clustering quality.

The framework of this study is presented in Section 2. The methods for driving behavior segmentation and segment clustering are described in Sections 3 and 4, respectively, followed by the results and discussion in Section 5. The findings of this paper provide valuable insights into the causes and effects underlying the generation of driving behaviors.

## 2. Materials and methods

### 2.1. Data description

Driving behavior primitives are defined as the minimal, physically meaningful segments generated by a driver when facing diverse driving scenarios, which include both static road environments and dynamic traffic conditions. Behavior primitives are influenced by the driving scenario, as well as the driver's experience and driving style. This study adopted a controlled variable approach to address the practical challenges of collecting a large and diverse dataset of driving behaviors, which would require data from a wide range of drivers across sufficiently rich scenarios. The driving scenario was fixed to a single environment, while the driver sample was deliberately recruited based on a preliminary demographic survey to encompass a range of experiences and driving styles. This method allows us to isolate and examine the influence of driver-specific factors on the resulting driving behavior primitives, thereby demonstrating the applicability and generalizability of our proposed extraction method.

The dataset for this study was collected using the RADS 8 DOF Panoramic Driving Simulator. The experimental route consists of 11 curves and coveres a total round-trip distance of approximately 21.7 km. The driving scenario included sparse surrounding traffic and pre-scripted traffic events triggered under specific conditions. A total of sixteen drivers (10 males, 6 females; age range: 28–50 years, mean = 29.8, SD = 2.7; driving experience: 0–12 years, mean = 7.6, SD = 3.3)

were recruited and compensated for their participation. All participants were instructed to drive under consistent conditions to minimize the influence of driving scenario.

Participants were recruited from the general public. The recruitment criteria required that all individuals hold a valid driver's license and have driven at least one hour per day over the past three months to ensure adequate driving familiarity and stable driving behavior. The study protocol was approved by the Ergonomics Research Ethics Committee of Transportation College, Jilin University. Prior to the experiment, all participants were fully informed of the research objectives, potential risks and benefits, and the intended usage of the collected driving data. Only driving behavior data is used in, no personally identifiable information is disclosed. Written informed consent was obtained from each participant.

The driving behavior data was sampled at a rate of 60 Hz. To reduce the volatility of the raw data, a moving-average algorithm was applied for smoothing. The statistical results from the processed data are shown in **Fig 1**. The velocity ranges from 0 to 34 m/s, longitudinal acceleration varies between $-7$ m/s$^2$ and 6 m/s$^2$, lateral acceleration ranges from $-7$ m/s$^2$ to 9 m/s$^2$, and longitudinal jerk spans from $-6$ m/s$^3$ to 6 m/s$^3$. Additionally, for the convenience of calculation and real-time analysis, the driving-straight and turning sections are determined based on the driving trajectory radius ($R$).

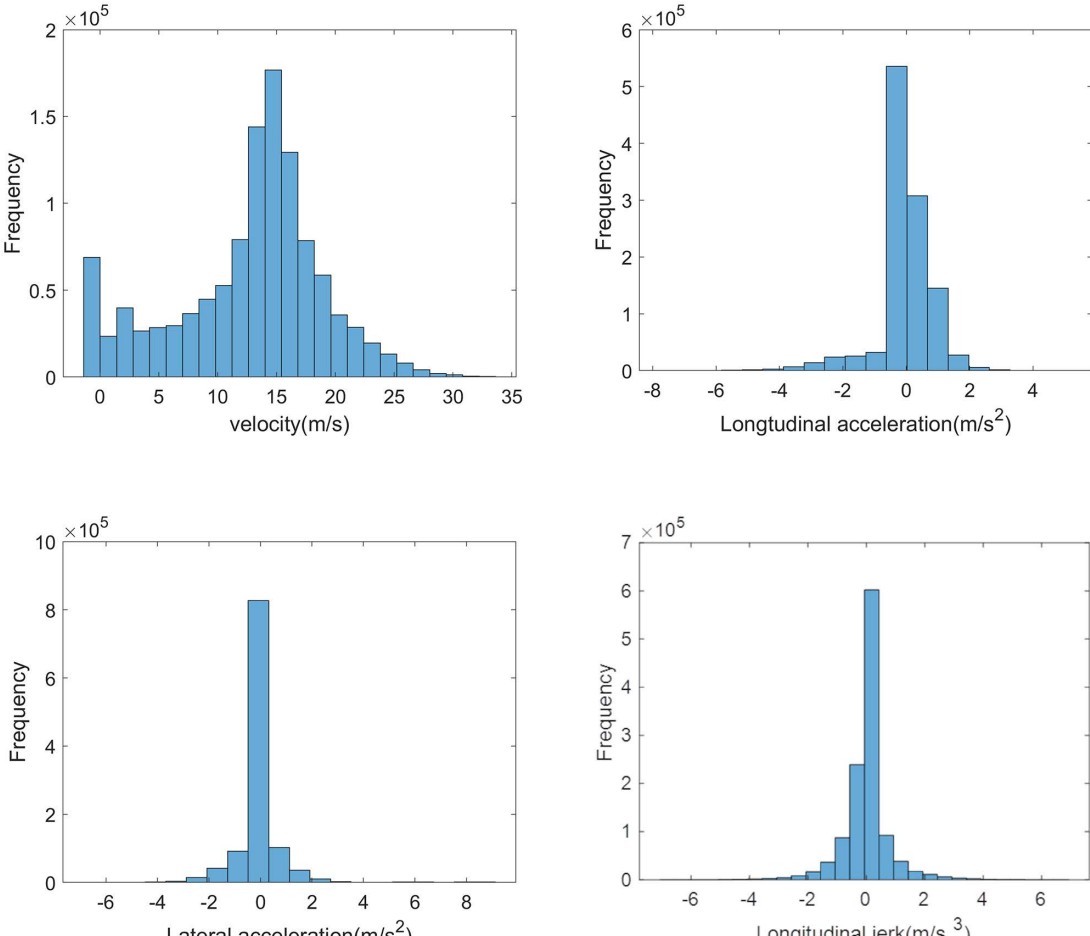

**Fig 1. The frequency distribution histograms of the key variables. (a)** velocity, **(b)** longitudinal acceleration, **(c)** lateral acceleration, **(d)** longitudinal jerk.

Specifically, when R > 1000 m, the vehicle is driving straight; when R < 1000 m, the vehicle is regarded as turning. Consequently, the following driving data pertain to either driving-straight or turning sections.

## 2.2. The research framework

Driving behavior primitives aim to enhance the understanding of complex driving behaviors by providing a thorough and fine-grained representation. To achieve this, the typical extraction process involves two core stages: driving behavior segmentation and segment clustering. The segmentation stage aims to partition continuous time-series driving data into individual segments, where the data trend within each segment remains stable, while adjacent segments exhibit different trends or characteristics. The clustering stage then groups these numerous segments into several clusters, ensuring that segments within the same cluster share similar features, whereas those in different clusters are distinct. Through this two-stage process, the resulting driving behavior primitives are finite in number, representing the fixed microscopic driving patterns formed by drivers based on their experience, and preferences.

In the segmentation stage, the selection of features is crucial as it directly determines the accuracy of the segmentation boundaries. To this end, in addition to using basic features that represent the vehicle motion state, we innovatively model the driver's subjective expectations on vehicle performance. This decision-level factor, which influences the control level, is constructed into a new feature set to significantly improve segmentation precision.

In the clustering stage, employing traditional feature engineering methods on a large number of segments with varying durations would inevitably lead to significant information loss. Therefore, topic models like LDA are typically adopted. However, the clustering performance of LDA is highly dependent on the discretization quality of driving behavior data. We propose a discretization method that considers non-linear coupling relationships among variables, providing a high-quality data foundation for LDA clustering.

Building on these key advancements in both the segmentation and clustering stages, we propose a novel framework for extracting driving behavior primitives, with the overall workflow illustrated in **Fig 2**.

## 3. Driving behavior segmentation based on the multi-type feature space

### 3.1. The BMASS for multi-dimensional time series segmentation

Driving behavior primitives are segments in which the vehicle's state or its change remains constant. Dividing the continuous driving behavior into independent segments in a reasonable way is of great significance.

As a data-driven segmentation algorithm, the Bayesian model-based agglomerative sequence segmentation (BMASS) algorithm has high robustness and is widely used in driving behavior segmentation. As a bottom-up, agglomerative sequence segmentation algorithm, it identifies the optimal split points to partition the sequence into meaningful segments, as illustrated in **Fig 3**. Each segment is modeled as a multivariate linear regression, where time serves as the independent variable, which is.

$$y_i = A_0 + At + Q^{\frac{1}{2}}\omega_i \tag{1}$$

The algorithm uses Bayesian methods to compute the marginal likelihood for each potential split point and evaluates the segmentation quality by comparing marginal likelihoods across different partitioning schemes. Starting from the finest initial state, where each data point represents a separate segment, the algorithm iteratively merges adjacent segments. For each merge, the marginal likelihood of the merged segment is recalculated, and the merge that maximizes the marginal likelihood is selected. This iterative process continues until a predefined stopping criterion is met. By balancing model complexity with data fit, optimal segmentation results could be obtained [11,18].

Existing literature typically relies on vehicle motion features to segment time-series driving behavior, as driving stability and safety are reflected in these explicit motion states. However, driving maneuvers are continually adjusted by drivers to

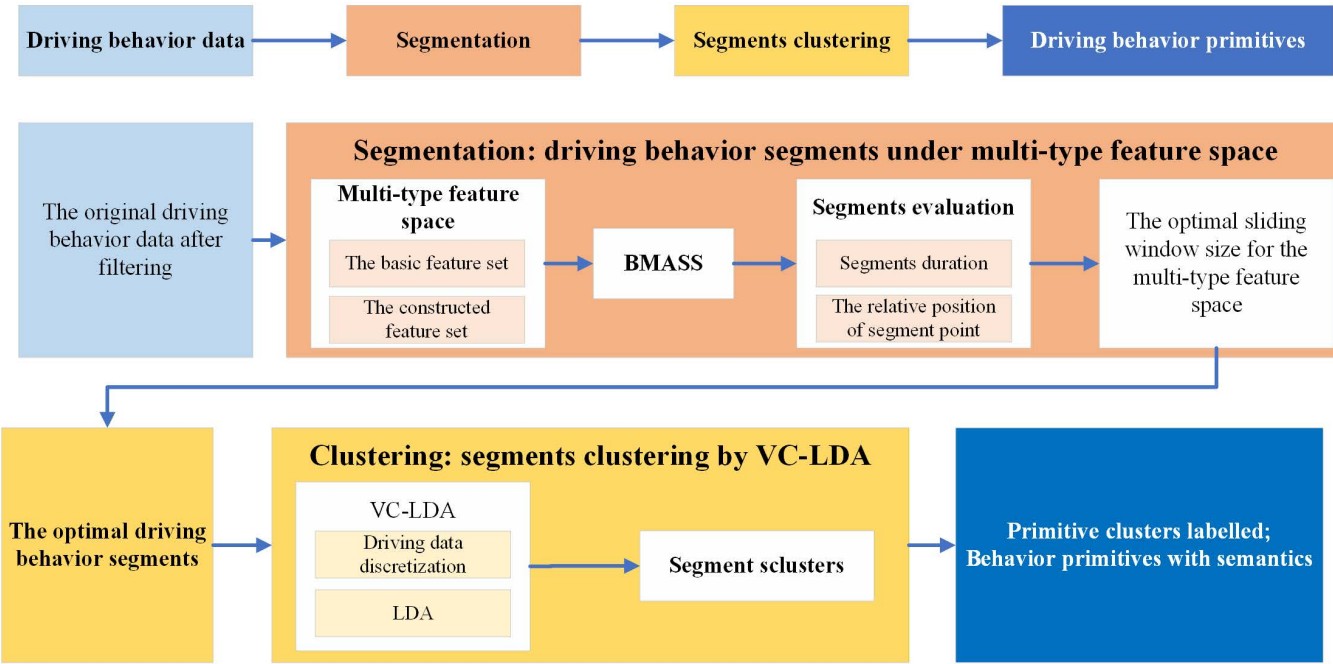

**Fig 2. The proposed framework for driving behavior primitive extraction.**

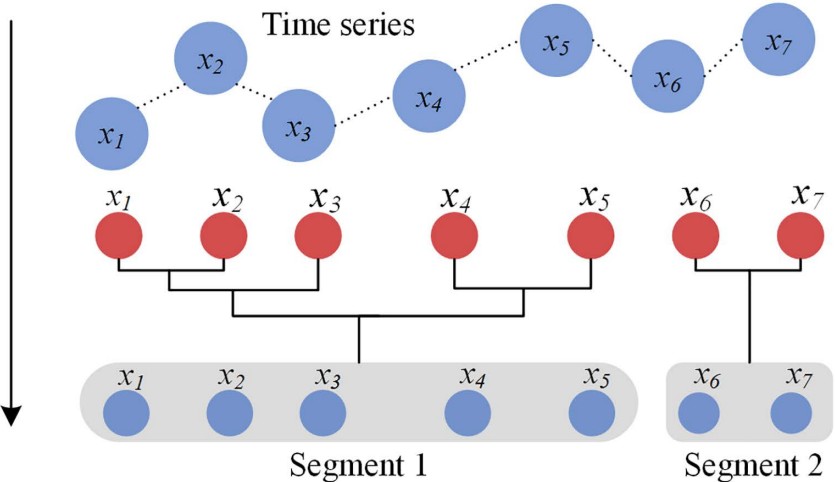

**Fig 3. Schematic diagram for series segmentation by BMASS.**

align with their personal expectations on vehicle driving performance. This implicit driver intent serves as the internal force guiding how a driver maneuvers the vehicle in response to the traffic environment.

In [28], the driving behavior model is structured into three hierarchical levels, as illustrated in **Fig 4**. At the strategic level, drivers establish overall trip objectives, such as minimizing travel time or distance, and plan the global route accordingly. At the tactical (or maneuvering) level, they make real-time decisions to balance multiple performance targets, such as safety, economy, and efficiency, while adapting to the evolving traffic conditions. At the control level, drivers execute

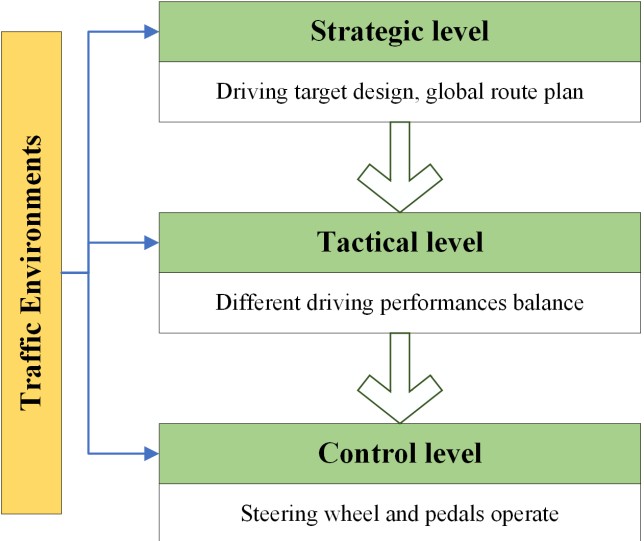

**Fig 4. The schematic diagram of the hierarchical driving behavior model.**

precise actions through the steering wheel and pedals to maintain vehicle control. Within this model, the tactical level involves short-term decision-making, while the control level deals with transient actions. Both are essential for a comprehensive analysis of driving behavior. Moreover, in the real-time motion planning and control of autonomous vehicles, it is common practice to design and dynamically weigh various performance objectives to achieve human-like driving [29,30].

Therefore, it is essential to incorporate features that convey these implicit intentions or expectations. To this end, a multi-type feature space should be constructed, combining a basic feature set that describes the vehicle motion state with a constructed feature set that represents the driver's latent expectations on vehicle driving performance. Driving behavior segmentation can then be performed using the BMASS method based on this enriched feature space.

### 3.2. Multi-type feature space

**3.2.1. The basic feature set.** Driving behavior refers to the quantifiable dynamic characteristics and movement patterns exhibited by a vehicle under certain traffic environment, which directly reflect the driver's control strategy, operational habits, and real-time intentions. Metrics such as velocity, acceleration, yaw rate, and body roll angle are often used to describe driving behavior from multiple dimensions. For passenger cars, which generally exhibit good driving stability, their behavior can be described in terms of longitudinal and lateral dimensions. Longitudinal driving behavior is typically characterized by longitudinal velocity and longitudinal acceleration, while lateral driving behavior is generally described using lateral acceleration. In addition, changes in longitudinal acceleration, also known as longitudinal jerk, are beneficial for driving behavior analysis [31].

In summary, velocity ($V$), longitudinal acceleration ($a_x$), lateral acceleration ($a_y$), and longitudinal jerk ($jerk$) are selected as the basic features for segmenting time-series driving behavior.

**3.2.2. The constructed feature set.** Drivers typically operate vehicles based on their expectations of driving performance, aiming to align vehicle motion states with their subjective preferences. Generally, driver expectation is a composite metric, representing a weighted combination of safety, efficiency, comfort, and fuel economy. Therefore, to obtain more realistic driving behavior segments, the constructed features must effectively describe these subjective

expectations. Following the study in [32], these features are derived from the integrals of quadratic terms of the vehicle's motion states, which is

$$x_{wT} = \int_0^T (x_n - x_e)^2 \, dt$$

(2)

where $x_n$=[v, ax, ay, jerk], and $x_{wT}$ respectively describes rapidity, fuel economy, safety, and comfort performance. $T$ is the sliding window size, and $x_e$ is the expectation value of each performance. Set expectation values of $a_x$, $a_y$ and *jerk* to zero, and the expectation value of *v* to the road speed limit. In this study, the speed limit is 22.2 m/s.

Thus, the constructed feature set is obtained using performance metrics for safety, efficiency, comfort, and fuel economy evaluated over a driving period.

### 3.2.3. Optimization of the sliding window size.

The constructed features are calculated by integrating over a specific time interval $T$, referred to as the sliding window size. To ensure temporal consistency between the basic and constructed feature sets, a moving average with the same window size $T$ is applied to the basic features, which is

$$x_{sT} = \frac{1}{N_T} \sum_{n=1}^{N_T} x_n$$

(3)

where $x_n$=[v, $a_x$, $a_y$, jerk], and $x_{sT}$ is the processed basic features. $T$ is the sliding window size, and $N_T$ is the number of data points in the sliding window.

The sliding window size is a key parameter that defines the resolution of the multi-type feature space and, consequently, influences the quality of behavior segmentation. Therefore, to identify the optimal window size, the segmentation quality across a range of window sizes is analyzed to select the one that yields the best performance.

(1) **Feasible sizes of the sliding window**

In practice, determining the optimal sliding window size through a traversal strategy is challenging. Therefore, the feasible sliding window sizes are first established based on previous studies to ensure the validity of the driving behavior primitive extraction.

Existing studies in intelligent vehicles and driving behavior modeling commonly analyze behaviors within a 1–4 second window. Accordingly, we set the upper bound to 4 seconds. To thoroughly investigate the parameter's impact on segmentation, we extend the lower bound to 0.5 seconds. This results in a final set of feasible window sizes, ranging from 0.5 to 4 seconds in 0.5s increments. The feasible sliding window sizes are T = [0.5, 1, 1.5, 2, 2.5, 3, 3.5, 4] s.

(2) **The optimal sliding window size**

Given the feasible sliding window size, the optimal sliding window size is selected by the data segmentation quality.

The segmentation quality is assessed based on the relative positions of segmentation points and the durations of the segments. The relative position of the segmentation point is,

$$p_i = x_i / l$$

(4)

where $p_i$ is the relative of the *i*-th segmentation point.

As shown in **Fig 5**, $x_i$ is the distance of segmentation point *i* between the start point of driving data to be segmented; *l* is the total length of driving data to be segmented.

The segment duration is,

$$d = t_1 - t_0$$

(5)

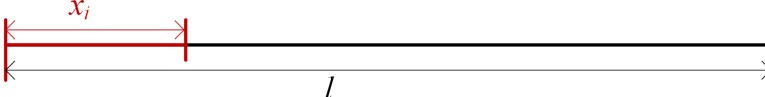

**Fig 5. Sketch map for the relative position of the *i*th segmentation point.**

where $d$ is the segment duration, $t_1$ is the end timestamp of the segment, and the $t_0$ is the start timestamp of the segment.

Driving behavior primitives represent the simplest forms of driving behavior, so the segment lengths are relatively short. Moreover, in the experiment presented in this paper, the density distribution of driving behavior is not predetermined. As a result, the segmentation points follow a uniform distribution throughout the entire time-series. In summary, shorter segment durations and more evenly distributed segmentation points indicate better segmentation results. Based on this, the optimal sliding window size is determined.

## 4. Behavior segments clustering based on VC-LDA

### 4.1. The VC-LDA

The segmentation stage divides continuous driving behavior into individual behavioral segments, where adjacent segments exhibit distinct characteristics and each segment represents a consistent driving mode. The semantic interpretation and categorization of these segments are then determined in the subsequent clustering stage.

Owing to the structural similarities between driving behavior and natural language, extended Latent Dirichlet Allocation (LDA) has been widely used to cluster segments into different categories to extract driving behavior primitives. However, LDA is designed to handle discrete text data, making it unsuitable for continuous time-series. Therefore, prior to clustering, the continuous driving behavior must first be discretized. The Gaussian Mixture Model (GMM) is a common choice for this task. GMM uses Gaussian distribution functions to model the probability of data occurrence, is applicable to multidimensional data, and can indirectly capture coupling relationships between variables through its covariance matrix [33].

However, GMM has fundamental limitations when applied to the complex multi-type feature space. Firstly, GMM is essentially a static model that treats each time-point's data as an independent sample, rendering it incapable of capturing the temporal asynchrony between variables. Secondly, GMM can only capture linear coupling relationships via its covariance matrix and fails to learn the complex nonlinear functional relationships among variables.

In the multi-type feature space, these complex couplings are particularly pronounced. On one hand, the coupling relationship among variables within the basic feature set (velocity, longitudinal acceleration, jerk, and lateral acceleration) can be uniformly described by a nonlinear 3-degree-of-freedom (3-DOF) vehicle dynamics model [30], which is inherently a system of nonlinear differential equations:

$$
\begin{aligned}
m(\dot{V}_x - V_y w) &= F_{xf}\cos\delta_f + F_{xr} - F_{yf}\sin\delta_f \\
m(\dot{V}_y + V_x w) &= F_{yf}\cos\delta_f + F_{yr} + F_{xf}\sin\delta_f \\
\dot{\psi} &= w \\
I_z\dot{w} &= aF_{yf}\cos\delta_f - bF_{yr} + aF_{xf}\sin\delta_f \\
X &= V_x\cos\psi - V_y\sin\psi \\
Y &= V_x\sin\psi + V_y\cos\psi
\end{aligned}
\tag{6}
$$

where $V_x$ is longitudinal velocity, $V_y$ is lateral velocity, $\psi$ is yaw angle, $w$ is yaw rate, $X$ is the longitudinal displacement in the inertial coordinate system, $Y$ is the lateral displacement in the inertial coordinate system, $m$ is the total mass of the vehicle, $\delta_f$ is the front wheel steering angle, $a$, $b$ are the distance from the vehicle's center of mass to the front axle and the rear axle respectively, $d$ is the wheelbase, $F_{xf}$, $F_{xr}$, $F_{yf}$, $F_{yr}$ are longitudinal and lateral force of front and rear tires, respectively.

The constructed features are derived from the basic features through integration, which inherently establishes a non-linear coupling relationship between the two feature sets. On the other hand, there exists temporal asynchrony between the basic feature set and the constructed feature set. The constructed features, obtained through methods like integration, reflect the driver's expectation of driving performance at the tactical level. This expectation precedes and influences the driver's control over the vehicle's motion, creating a time lag.

Therefore, GMM cannot adequately capture these nonlinear and time-lagged coupling relationships. Furthermore, as a purely data-driven approach, GMM's discretization can be adversely affected by abnormal data fluctuations, potentially leading to discretized states that lack clear physical meaning.

To address these limitations, we propose a discretization method based on vehicle dynamics. This method first analyzes the relationships between multiple features within the multi-type feature space from a vehicle dynamics perspective, and then performs the discretization. The Variable Coupling-based Latent Dirichlet Allocation (VC-LDA) method proposed in this paper is employed to cluster the segments, as illustrated in **Fig 6**. First, functions that describe the coupling relationships are derived based on vehicle dynamics. These functions are then used to calculate the discretization thresholds. Subsequently, LDA is applied to the segments, which are reconstructed from the discretized data, to perform the final clustering of behavior segments.

## 4.2. Driving behavior discretization under variables coupling analysis

Driving variable discretization is achieved by partitioning the four basic and four constructed features into distinct intervals. This process transforms the continuous time-series driving behavior into a sequence of discrete symbols. However, a simple, independent discretization is insufficient due to inherent coupling relationships among the variables, which are dictated by vehicle dynamics. For instance, a vehicle's acceleration capability diminishes as its velocity increases [34]. Consequently, it is crucial to define these discretization intervals by explicitly accounting for these inter-variable couplings.

A prime example of this coupling is the relationship between longitudinal acceleration and vehicle velocity. As illustrated in **Fig 7**, the observed longitudinal acceleration exhibits a decreasing trend with increasing velocity. Therefore, the upper and lower envelopes of longitudinal acceleration can be effectively modeled as a function of velocity, expressed as follows:

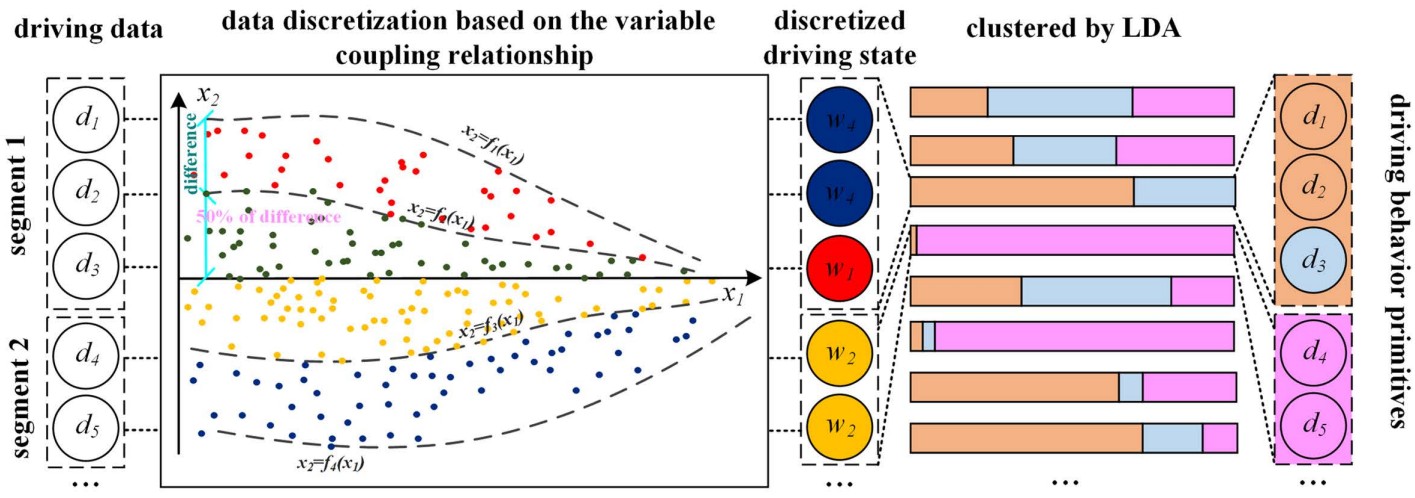

**Fig 6. The Segments clustering process by VC-LDA.**

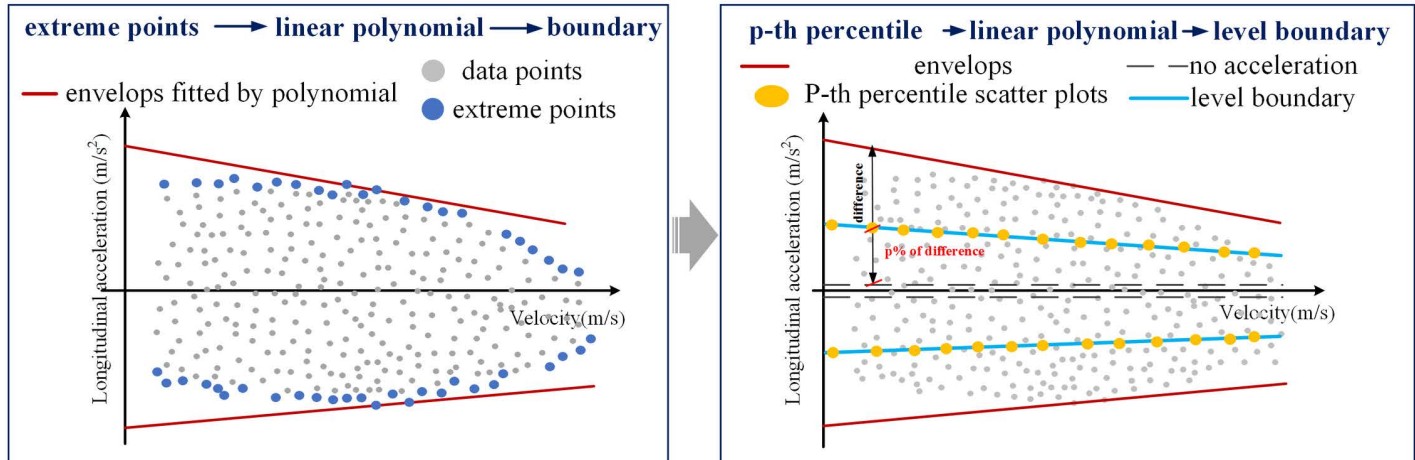

**Fig 7. Interval division of driving behavior variables considering the variable coupling relationship.**

$$\begin{cases} y_{upper} = a_{1n}x^n + a_{1(n-1)}x^{n-1} + ... + a_{10} \\ y_{lower} = a_{2n}x^n + a_{2(n-1)}x^{n-1} + ... + a_{20} \end{cases}$$

(7)

where $y_{upper}$ and $y_{lower}$ are the upper and lower envelops of longitudinal acceleration, $x$ is vehicle velocity, $a_{1n}, a_{1(n-1)}, ..., a_{10}$ are coefficients of the upper envelop fitted by the upper limits of the scatter plot; $a_{2n}, a_{2(n-1)}, ..., a_{20}$ are coefficients of the lower envelop fitted by the bottom limits of the scatter plot.

To discretize longitudinal acceleration into distinct driving states, we employ a percentile-based adaptive thresholds approach that accounts for the velocity-dependent nature of acceleration capability. This method establishes adaptive boundaries that vary as functions of vehicle velocity, rather than relying on fixed acceleration thresholds. These boundaries, which separate distinct acceleration levels, are derived from the percentile distribution of acceleration values at each given velocity [30].

The maximum acceleration and deceleration capabilities at a given velocity can be determined using Equations (7). As illustrated in **Fig 7**, an acceleration level boundary is calculated by fitting the $P$-th percentile values of acceleration across different vehicle speeds. This is expressed as:

$$y_{level\ boundary} = b_n x^n + b_{n-1} x^{n-1} + ... + b_0$$

(8)

where $b_n, b_{n-1}, ...b_0$ are the coefficients of level boundary fitted by $P$-th percentile values at each velocity point.

By combining these fitted level boundaries with the upper and lower envelopes, longitudinal acceleration can be discretized into several levels, effectively accounting for its coupling relationship with driving velocity. This method offers several key advantages: it adapts to the non-linear relationship between acceleration capability and velocity, accounts for varying data density across different velocity ranges, and provides statistically meaningful boundaries that reflect the actual distribution of driving behaviors rather than arbitrary fixed values. Consequently, this approach ensures that the discretization boundaries are physically meaningful and contextually appropriate, maintaining consistency with the underlying vehicle dynamics constraints observed in the data.

The same methodology can be applied to the interval division of other variables. For each variable, the process begins by identifying its most significant influencing factor and the associated coupling relationships. Subsequently, the variable's

upper and lower envelopes, along with its discretization level boundaries, are fitted. Finally, based on the interval division of all eight variables, the discretization of driving behavior is performed.

## 5. Results and discussion

### 5.1. The optimal segmentation results

The multi-type feature space using various sliding window sizes is calculated and subsequently input these spaces into the BMASS algorithm to generate corresponding segmentation results.

The statistical characteristics of segment durations for each sliding window sizes are shown in **Figs 8 and 9**. A key observation is that the 1- and 4-second sliding window yield a higher proportion of short segments (those under 6 seconds). This suggests that the 1- and 4-second sliding windows are more sensitive to rapid changes in driving behavior.

**Fig 10** illustrates the distribution of the relative positions of segmentation points for various sliding window sizes, with all curves exhibiting a bell-shaped pattern. To identify the optimal sliding window size, we quantitatively analyzed the width and flatness of the top region for each curve.

The top region was defined using a peak-percentage threshold method. Specifically, all points where the probability density is greater than 70% of the peak value $P_{max}$ are considered part of the top region. The coverage of this region along the horizontal axis is defined as the top region width. The flatness of the top region is used to quantify its volatility, which is

$$\begin{cases} S = 1/(1 + f_r) \\ f_r = (P_{max} - T_{min})/T_{mean} \end{cases} \tag{9}$$

where $T_{min}$ and $T_{max}$ are the minimum and maximum value in the top-region.

The analysis of the eight curves in **Fig 10** reveals that the distribution curve for the 1-second sliding window achieves the optimal top-region flatness. Concurrently, its top-region width, while slightly smaller than that of the 0.5-second window, is larger than that of all other curves.

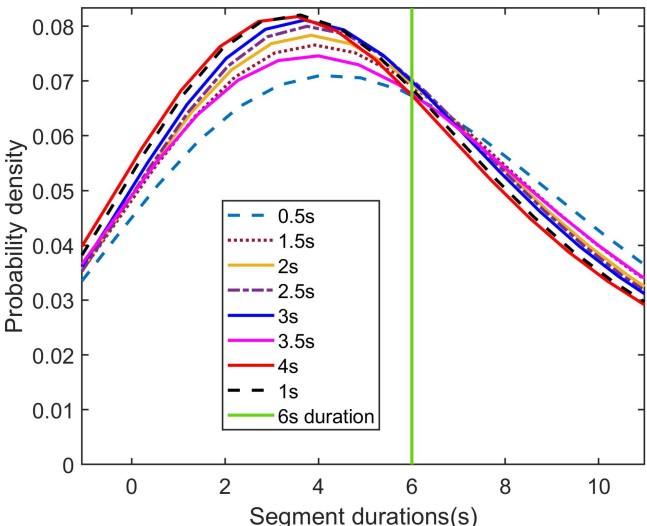

**Fig 8. Probability density distributions of segment durations for different sliding window sizes.**

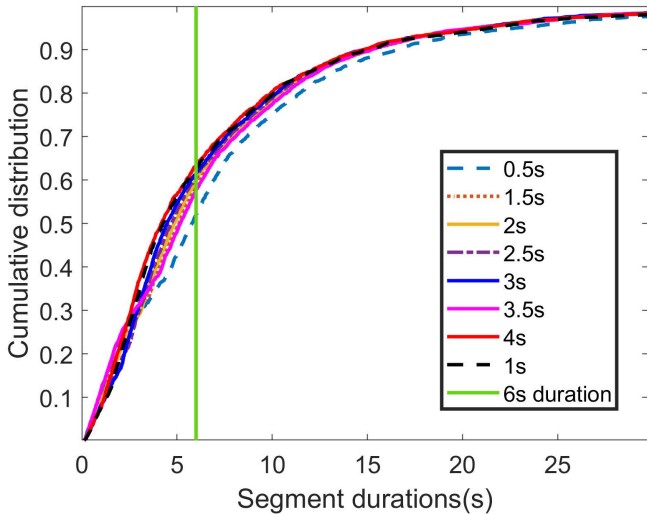

**Fig 9. The cumulative distribution of segment durations for different sliding window sizes.**

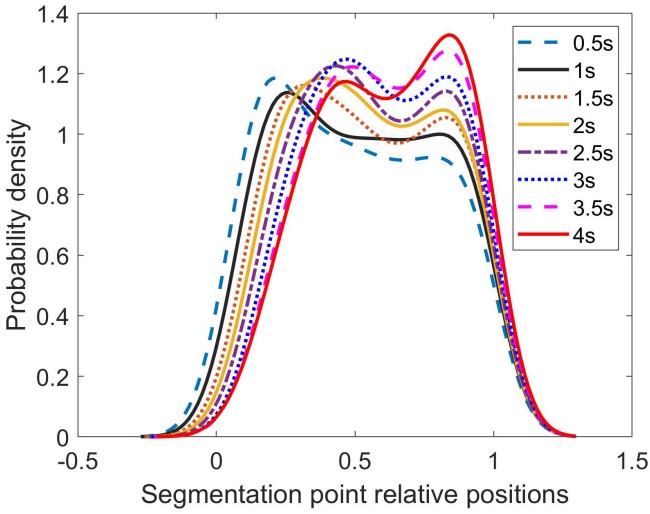

**Fig 10. Probability density of the relative positions of segmentation points across different sliding window sizes.**

Considering both the distribution of segment durations and the relative positions of segmentation points, we conclude that a 1-second sliding window is optimal. This window size effectively captures changes in driving behavior while producing a reasonable segmentation. Therefore, the 1-second window is selected for generating the optimal segments.

Applying the BMASS algorithm to the multi-type feature space calculated with the 1-second window yielded a total of 2,957 segments. **Fig 11** presents a representative segmentation result for a specific turning maneuver. Vertical dotted lines denote the segmentation points, with the interval between two consecutive points representing a single driving behavior primitive. The visualization confirms that the variable profiles differ significantly between adjacent segments and that the segmentation points are uniformly distributed throughout the time series.

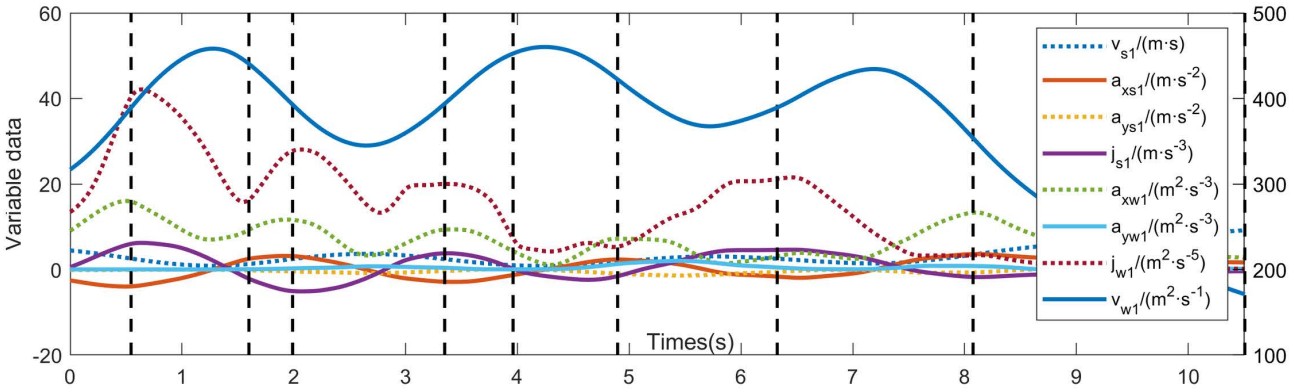

**Fig 11. Behavior segmentation result on a turning section.**

## 5.2. Segments clustering results

### 5.2.1. Driving behavior discretization.

(1) **Velocity discretization**

Vehicle driving velocity is an independent variable and is commonly discretized into three levels: low, medium, and high [35]. The discretization is determined by two velocity thresholds: 30 km/h and 60 km/h. Therefore, velocity locates in $[0, 30)$ km/h, $[30, 60)$ km/h, and $[60, V_{max}]$ km/h are defined as low, medium and high speed, respectively. $V_{max}$ is the maximum velocity in the simulator experiment.

(2) **Jerk discretization**

Jerk is defined as the rate of change of longitudinal acceleration. Its discretization is often based on its correlation with driver subjective perception, typically categorizing it into gentle and aggressive levels [36].

Following the methodology in [36], a threshold of 3.6 m/s³ is used to distinguish between these levels. Based on the established threshold of 3.6 m/s³, positive jerk values greater than this limit are classified as aggressive, while those between 0 and 3.6 m/s³ are considered gentle. Similarly, for negative jerk, values below −3.6 m/s³ are classified as aggressive, and those between −3.6 m/s³ and 0 are considered gentle.

(3) **Longitudinal acceleration discretization**

In accordance with vehicle dynamics, longitudinal acceleration decreases as velocity increases. To capture this relationship while maintaining model simplicity, the upper and lower envelopes of longitudinal acceleration are modeled using a first-order polynomial (i.e., a linear function). This linear approximation provides a simplified yet effective representation of the acceleration's dependency on velocity. Based on the empirical driving data shown in **Fig 12**, the fitted envelope functions are

$$\begin{cases} a_{x,max}(V) = -0.09V + 4.11 \\ a_{x,min}(V) = 0.14 \cdot V - 8.00 \end{cases} \tag{10}$$

To discretize the acceleration, we adopt a five-level classification system established in prior research: no acceleration, gentle acceleration, aggressive acceleration, gentle deceleration, and aggressive deceleration [37]. The "no acceleration" state, which characterizes smooth driving, is defined by the interval [−0.05, 0.05] m/s² [37].

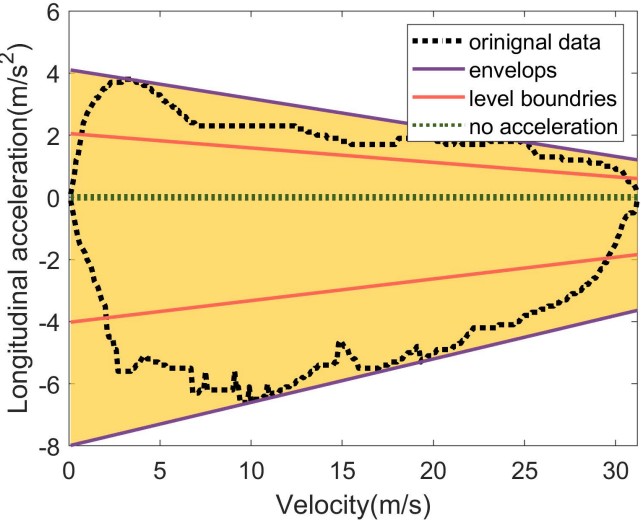

**Fig 12. Longitudinal acceleration discretiazation considering its coupling relationships with velocity.**

With the "no acceleration" interval established, the remaining boundaries are determined as follows. The boundary separating gentle from aggressive acceleration, as well as the boundary separating gentle from aggressive deceleration, is derived by fitting a function to a specific percentile of the acceleration data. This boundary is fitted using the 50th percentile of the acceleration values at each velocity, effectively representing the transition threshold between gentle and aggressive driving behavior. The boundaries are expressed as

$$\begin{cases} a_{x,50\%+}(V) = -0.05V + 2.05 \\ a_{x,50\%-}(V) = 0.07V - 4.02 \end{cases} \tag{11}$$

where $P$-th percentile is set as median to represent the boundary of gentle acceleration and aggressive acceleration.

## (4) Lateral acceleration discretization

Lateral and longitudinal accelerations are fundamentally coupled due to the performance limits of the tires, a relationship often described by the friction circle concept. This coupling can be mathematically represented by

$$a_y(a_x) = \sqrt{a_R^2 - a_x^2} \tag{12}$$

where $a_R$ is the maximum absolute value of longitudinal and lateral acceleration in the dataset.

From the experimental data, $a_R$ is determined to be 6.5 m/s². Consequently, the absolute physical boundary for lateral acceleration, as shown in **Fig 13**, can be modeled by

$$a_y(a_x) = \sqrt{6.5^2 - a_x^2} \tag{13}$$

For the purpose of discretization, lateral acceleration is typically divided into two levels: slight lateral sideslip and significant lateral sideslip. The boundary that separates these two driving states was established based on the percentile-based method referenced earlier. By setting the percentile value to 50% (the median), the level boundary is derived as follows

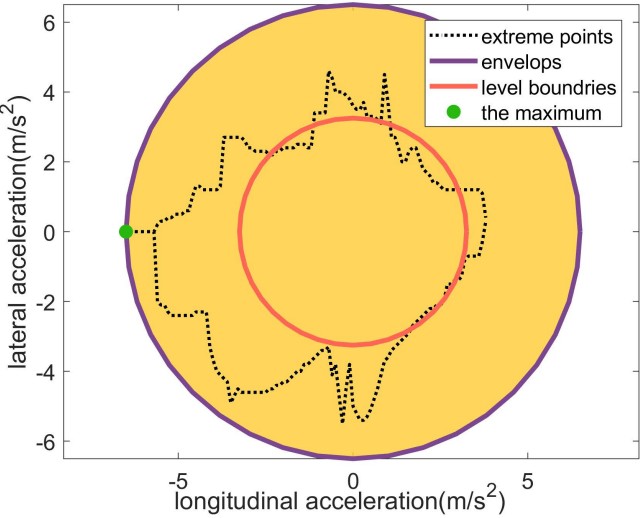

**Fig 13. Lateral acceleration discretization considering its coupling relationships with lonitudinal acceleration.**

$$a_{y,50\%}(a_x) = \sqrt{3.25^2 - a_x{}^2} \tag{14}$$

## (5) The rapidity performance discretization

As another key performance indicator derived from driving velocity, the vehicle's rapidity performance is also discretized. Based on the established velocity intervals, the rapidity performance can be classified into three levels: good, moderate, and poor.

The thresholds for these rapidity levels are determined by analyzing the empirical driving data within each velocity interval. This results in the following rapidity intervals, $[0, 30)$ m²/s, $[30, 190)$ m²/s, $[190, V_{w,max})$ m²/s.

## (6) The comfort performance discretization

Similarly, driving comfort performance is classified into two levels, good and poor, based on the jerk intervals. The threshold separating these levels is determined to be 12.8 m²/s⁵, which is calculated using the same percentile-based methodology as the jerk level boundary.

## (7) The economy performance discretization

According to Equation (2), economy performance is modeled as a function of velocity squared. To characterize its upper limit, a second-order polynomial (quadratic function) is fitted to the empirical data, as shown in **Fig 14**. This fitted upper envelope is represented by

$$a_{xw,max}(V) = -0.11V^2 + 2.73V + 17.35 \tag{15}$$

For discretization, the economy performance is categorized into two levels: good and poor. The lower boundary is defined as zero. Consequently, the threshold separating good from poor performance is derived from the 50th percentile of the performance data, which is

$$a_{xw,50\%}(V) = -0.06V^2 + 1.56V + 14.11 \tag{16}$$

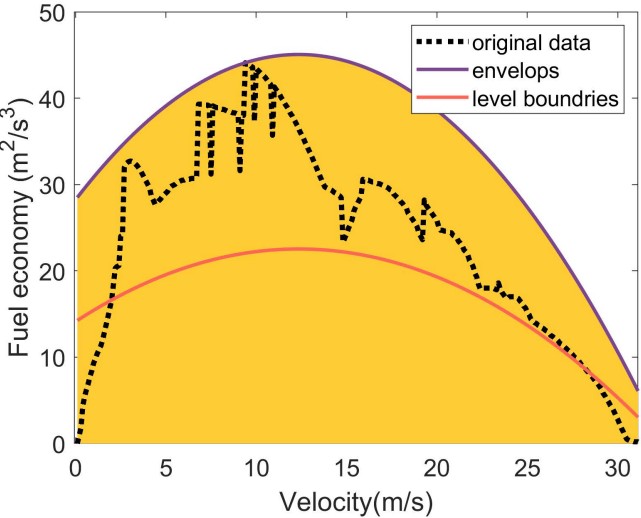

**Fig 14. The economy performance discretization considering its coupling relationship with velocity.**

### (8) The safety performance discretization

Similarly to comfort performance, the upper envelope of safety performance is obtained by fitting a function to the empirical dataset, as shown in **Fig 15**. This fitted envelope is represented by

$$a_{y,\max}(a_x) = -0.95a_x{}^2 - 2.70a_x + 19.76 \tag{17}$$

The safety performance is then discretized into two levels: good and poor. With the lower boundary defined as zero, the threshold separating these levels is calculated by the 50th percentile of the performance data, which is

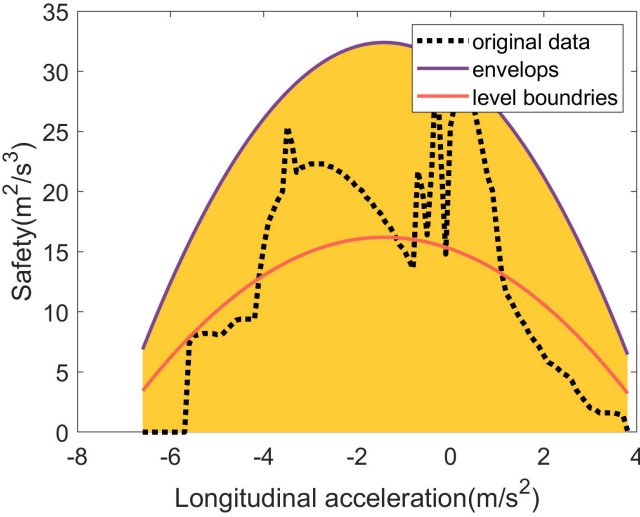

**Fig 15. The safety performance discretization considerting its coupling relationship with longitudinal acceleration.**

$$a_{yw,50\%}(a_x) = -0.48a_x{}^2 - 1.35a_x + 15.23 \qquad (18)$$

The interval boundaries detailed in **Table 1** are used to discretize each variable into a specific state. The driving behavior are then categorized by considering the comprehensive combination of all variable states. This process partitions the behavior into 166 bins, which is significantly less than the theoretical maximum of 2880. Each bin represents a unique driving state, defined by the vehicle's operational state and the driver's behavioral characteristics, as captured by both the basic and constructed feature sets.

**5.2.2. Segment clustering.** After discretizing the time-series driving behavior, the segments formed by the discretized symbols are input into the standard LDA for clustering. During the clustering process, the number of clusters ($K$) must be predetermined.

Drivers tend to rely on simple behavioral maneuvers in practical driving, so the number of primitive clusters is expected to be limited. Based on existing research, the number of primitive clusters ($K$) is set to range from 2 to 10. The optimal number of clusters is determined by evaluating perplexity. **Fig 16** presents the perplexity values for different numbers of clusters. The perplexity gradually decreases as the number of clusters increases. However, when the number of clusters exceeds 5, the reduction in perplexity fluctuates around 2%. This indicates that the clustering performance does not improve significantly with more than 5 clusters. Therefore, to strike a balance between computational efficiency and clustering effectiveness, we set the number of clusters to k = 5, yielding a total of five categories of driving behavior primitives. The statistical features of each cluster are presented in **Table 2**.

The physical meaning of each primitive is defined by integrating its statistical features, the prevailing environmental context, and the driver's primary task. A detailed interpretation is as follows:

**Cluster 1: Stable high-speed cruising**

*Observation:* This cluster is characterized by high mean velocity with low standard deviations in both velocity and longitudinal acceleration. The constructed variables for rapidity and comfort also exhibit low means and variances.

**Table 1. Driving variables discretization considering variables coupling relationships.**

| Variable | Discretization criteria | Levels | Variable | Discretization criteria | Levels |
|---|---|---|---|---|---|
| $V$ (km/h) | < 30 | Low velocity | $V_w$ (m²/s) | < 30 | Good rapidity |
| | 30~60 | Medium velocity | | 30~190 | Moderate rapidity |
| | > 60 | High velocity | | > 190 | Poor rapidity |
| $a_y$ (m/s³) | $\leq \sqrt{3.25^2 - a_x{}^2}$ | Slight lateral Displacement | $a_{yw}$ (m²/s³) | $\leq -0.48a_x{}^2 - 1.35a_x + 15.23$ | Good safety |
| | $> \sqrt{3.25^2 - a_x{}^2}$ | Significant lateral displacement | | $> -0.48a_x{}^2 - 1.35a_x + 15.23$ | Poor safety |
| $a_x$ (m/s²) | $> -0.05V + 2.05$ | Aggressive acceleration | $a_{xw}$ (m²/s³) | $\leq -0.06V^2 + 1.36V + 14.11$ | Good fuel economy |
| | $0.05 \sim -0.05V + 2.05$ | Gentle acceleration | | | |
| | $-0.05 \sim 0.05$ | No acceleration | | $> -0.06V^2 + 1.36V + 14.11$ | Poor fuel economy |
| | $-0.05 \sim 0.07V - 4.02$ | Gentle deceleration | | | |
| | $< 0.07V - 4.02$ | Aggressive deceleration | | | |
| jerk (m/s³) | > 3.6 | Aggressive positive jerking | $jekr_w$ (m²/s⁵) | $\leq 12.8$ | Good comfort |
| | 0~3.6 | Gentle positive jerking | | | |
| | 0~−3.6 | Gentle negative jerking | | $> 12.8$ | Poor comfort |
| | <−3.6 | Aggressive negative jerking | | | |

 

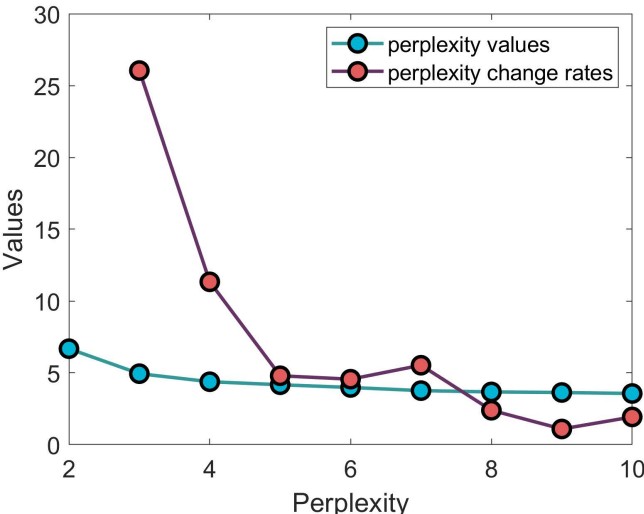

**Fig 16. Perplexity under different number of clusters.**

**Table 2. Statistical features of driving behavior primitive clusters.**

| | | V | $a_x$ | $a_y$ | jerk | $V_w$ | $a_{xw}$ | $a_{yw}$ | $jerk_w$ |
|---|---|---|---|---|---|---|---|---|---|
| **Mean** | Cluster 1 | 19.49 | 0.02 | −0.03 | −0.13 | 17.30 | 0.40 | 0.61 | 0.52 |
| | Cluster 2 | 13.44 | 0.26 | −0.05 | −0.09 | 82.22 | 0.36 | 0.79 | 0.55 |
| | Cluster 3 | 4.85 | 0.29 | −0.04 | 0.09 | 315.30 | 2.02 | 0.50 | 2.37 |
| | Cluster 4 | 6.67 | −0.40 | −0.01 | 0.20 | 266.88 | 1.41 | 0.23 | 1.54 |
| | Cluster 5 | 13.70 | −0.24 | −0.03 | 0.10 | 77.29 | 0.56 | 0.45 | 0.66 |
| **Standard deviation** | Cluster 1 | 3.13 | 0.61 | 0.76 | 0.64 | 16.10 | 1.02 | 1.90 | 1.64 |
| | Cluster 2 | 2.21 | 0.51 | 0.87 | 0.68 | 41.93 | 0.58 | 1.42 | 1.76 |
| | Cluster 3 | 3.52 | 1.30 | 0.67 | 1.36 | 107.86 | 3.90 | 1.71 | 5.13 |
| | Cluster 4 | 4.90 | 1.04 | 0.47 | 1.07 | 108.02 | 3.21 | 0.94 | 2.74 |
| | Cluster 5 | 2.12 | 0.66 | 0.66 | 0.73 | 41.11 | 1.62 | 1.06 | 1.86 |

*Interpretation:* This profile is predominantly observed during straight-line driving on open roads. The driver's primary task is to maintain a steady speed efficiently. The minimal fluctuations in acceleration and jerk indicate a non-demanding, predictable driving scenario where the primary goal is efficient travel. *Semantic Definition:* Therefore, this primitive is defined as "Stable High-Speed Cruising", representing a state of efficient and comfortable travel in predictable traffic conditions.

### Cluster 2: Cautious lane-changing/turning

*Observation:* This cluster exhibits high mean and standard deviation of lateral acceleration, yet low values for longitudinal acceleration and fuel consumption. *Interpretation:* The significant lateral acceleration indicates a maneuver involving a change in direction, such as a lane change or a turn. The gentle longitudinal acceleration suggests the driver is executing this maneuver deliberately and smoothly, rather than reactively. This points to a planned, cautious action aimed at maintaining safety and fuel efficiency. *Semantic Definition:* This primitive is defined as "Cautious Lane-Changing/Turning", representing a planned lateral maneuver with a focus on smoothness and fuel economy.

### Cluster 3: Aggressive stop-and-go navigation

*Observation:* This cluster shows low mean velocity but high means and standard deviations for velocity, longitudinal acceleration, and jerk. The high standard deviation of jerk is particularly notable. *Interpretation:* This pattern is characteristic of vehicle start-up from a standstill, often in congested traffic. The intense and fluctuating acceleration/deceleration suggests the driver is reacting impatiently or aggressively to the surrounding traffic flow, trying to gain a positional advantage. *Semantic Definition:* This is defined as "Aggressive Stop-and-Go Navigation", a behavior characterized by harsh control inputs and poor performance, reflecting an impatient driving style.

### Cluster 4: Reactive emergency braking

*Observation:* This cluster features a lower mean velocity with high variability. The longitudinal acceleration shows large negative mean values, and jerk is also high. *Interpretation:* The strong negative acceleration (braking) and high jerk are indicative of a sudden, forceful deceleration. This is a reactive, safety-critical maneuver, almost certainly triggered by an unexpected event, such as a sudden stop of the vehicle ahead or an emerging hazard. *Semantic Definition:* This primitive is semantically labeled "Reactive Emergency Braking", signifying a high-priority, safety-critical response to an imminent danger.

### Cluster 5: Normal urban/suburban driving

*Observation:* All variables for this cluster are at moderate levels, with the exception of a low standard deviation for velocity. *Interpretation:* This cluster represents a baseline or default driving state, typical of city or suburban roads with light to moderate traffic. The moderate speeds and accelerations, combined with stable velocity, suggest the driver is maintaining a vigilant but relaxed cruise, making minor adjustments to the traffic flow. *Semantic Definition:* This is defined as "Normal Urban/Suburban Driving", representing a neutral state of vehicle operation focused on maintaining safe following distances and smooth speed regulation.

### 5.3. Superiority of the multi-type feature space

Driving behaviors are segmented by BMASS using two distinct feature spaces: the traditional basic feature space and our proposed multi-type feature space. The evaluation on segmentation was based on the relative positions of segmentation points and the durations of the resulting segments.

As shown in **Figs 17** and **18**, the traditional basic feature space produces a significant number of suboptimal segments, including many that are excessively long (>12 seconds) or overly short (<1 second). This indicates that the segmentation is either too coarse or too granular.

Moreover, an analysis of the distribution of segmentation points (**Fig 19**) shows that the traditional basic feature set produces a smaller top-region width and larger top-region flatness. In contrast to a more uniform distribution, this pattern signifies an over-representation of segmentation points at the data's onset, which often corresponds to incomplete or unstable segments.

In summary, the multi-type feature space demonstrably improves segmentation quality. It effectively reduces the occurrence of both overly long and overly short segments, while also minimizing the percentage of incomplete segments. This allows for the extraction of more precise segments that truly represent driving behavior primitives, thus highlighting the superiority of the multi-type feature space for this task.

### 5.4. Superiority of the VC-LDA clustering

To validate the proposed VC-LDA model, a benchmark comparison against the GMM-LDA model was conducted. Both models were tasked with clustering driving behavior primitives into five categories, with perplexity serving as the primary

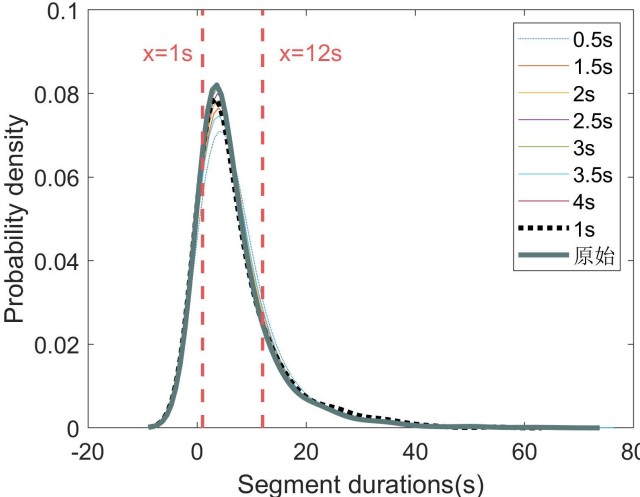

**Fig 17. Probability density distributions of segment durations under different spaces.**

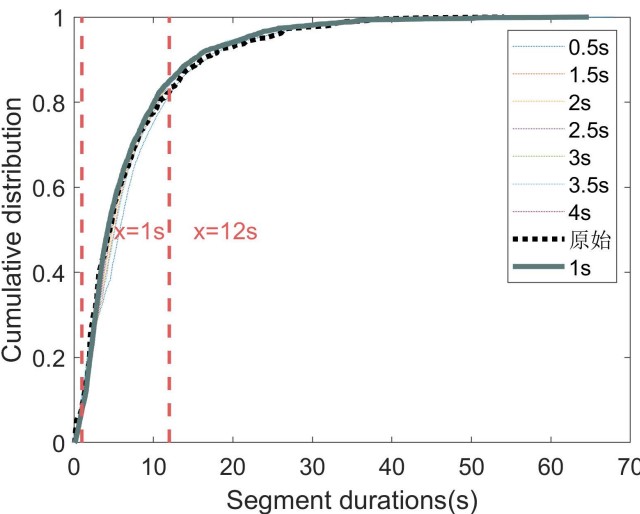

**Fig 18. The cumulative distribution of segement durations under different spaces.**

evaluation metric. The VC-LDA model achieved a perplexity of 3.15, substantially lower than the 14.36 for GMM-LDA, demonstrating its superior overall clustering performance.

To delve deeper into why VC-LDA performs better, the internal structure of the clusters is visualized. Since LDA is a bag-of-words model, the multi-dimensional, time-series driving data is discretized into 166 unique bins, which function as the model's words. A radar chart (**Figs 20** and **21**) to visualize the topic-word distributions is then used, where each "topic" corresponds to a primitive cluster.

In this visualization, each of the 166 rays represents a single "word" (bin). The length of a ray, extending from the center, indicates the normalized frequency of that "word" within a specific "topic" (primitive cluster). Each colored curve therefore plots the unique "word distribution" signature for its respective cluster.

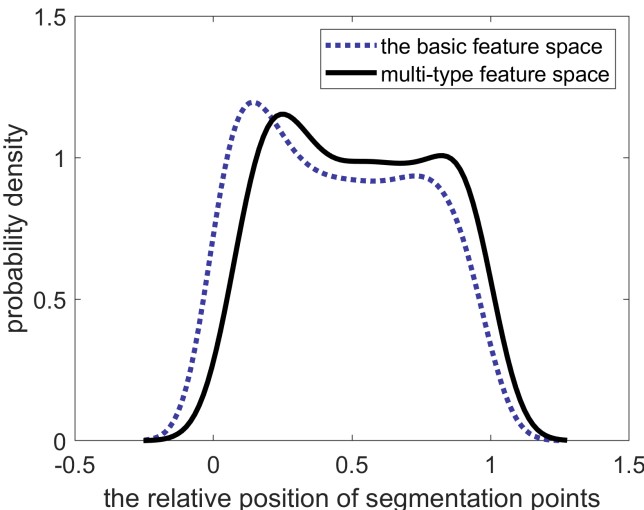

**Fig 19. Probability density of the relative positions of segmentation points under different spaces.**

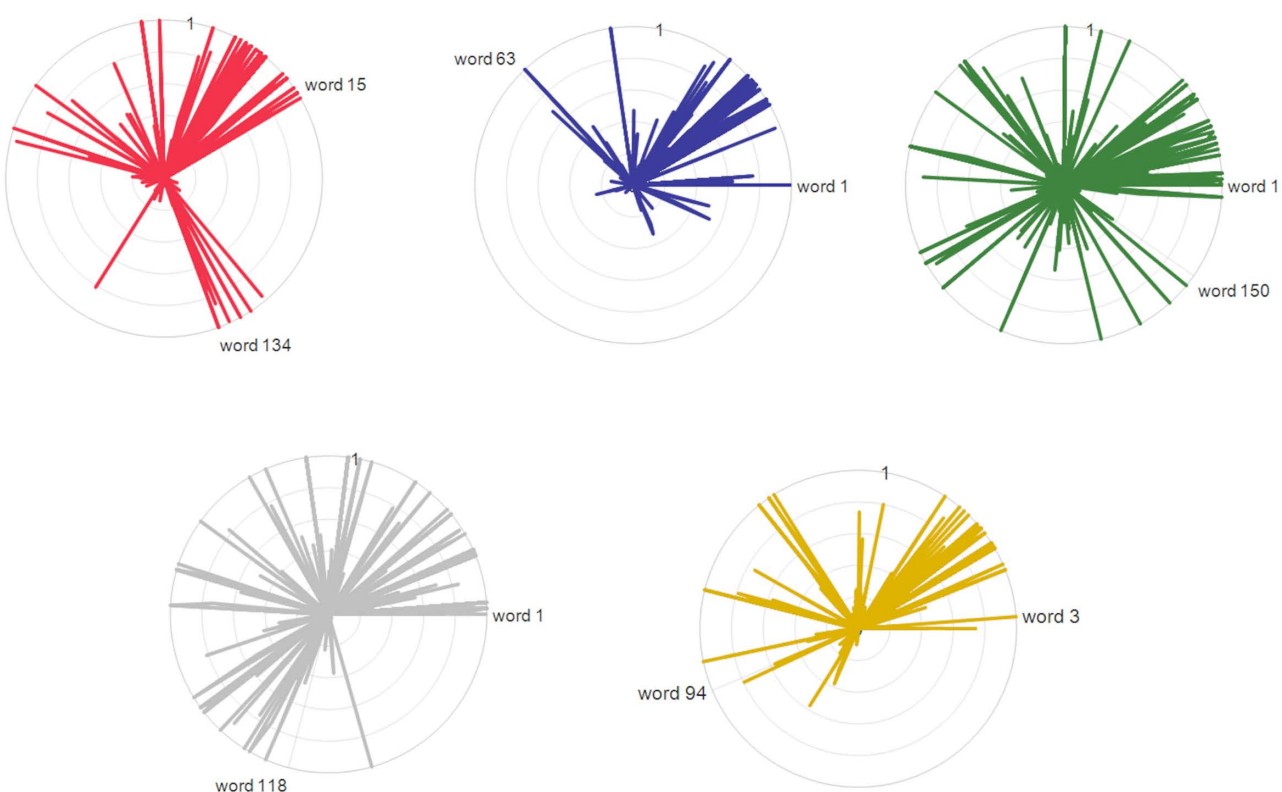

**Fig 20. Radar chart to visualize the topic-word distributions in primitives obtained by VCDA. (a)** Cluster 1: (bin 15～134), **(b)** Cluster 2: (bin 1～63), **(c)** Cluster 3: (bin 1～150), **(d)** Cluster 4: (bin 1～118), **(e)** Cluster 5: (bin 3～94).

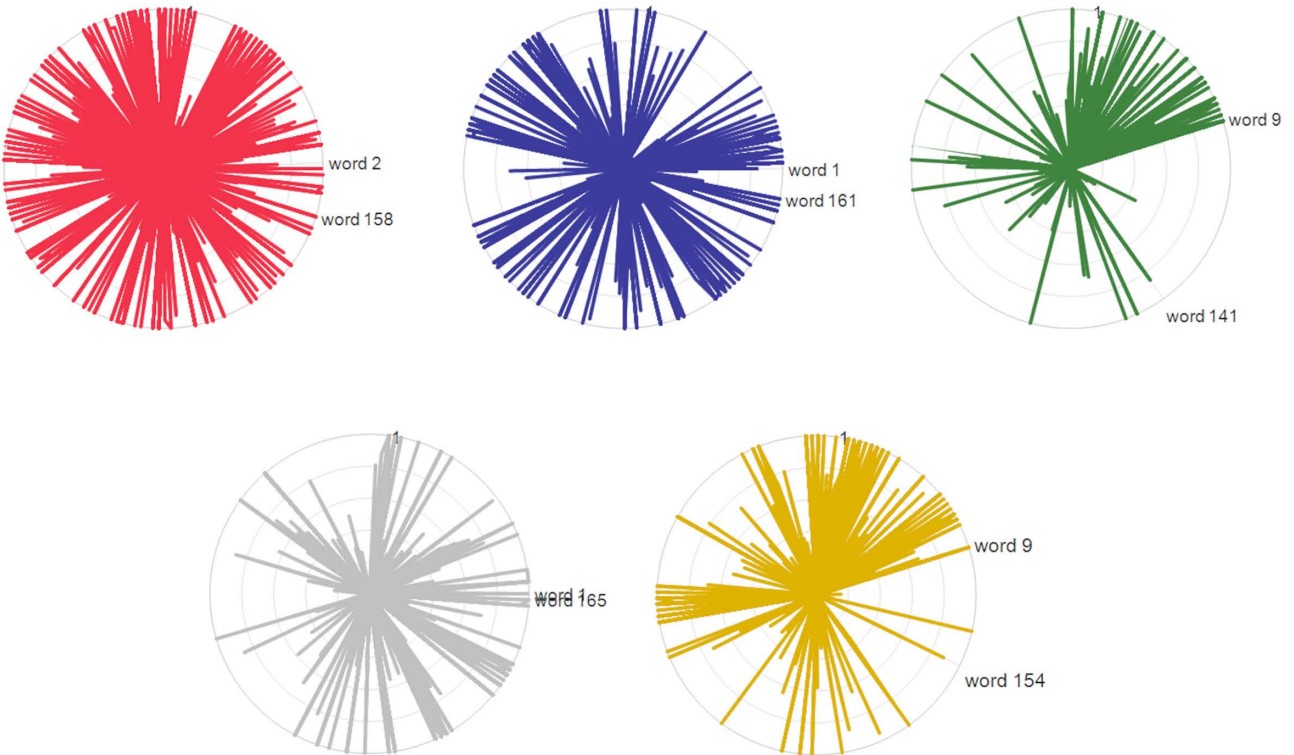

**Fig 21. Radar chart to visualize the topic-word distributions in primitives obtained by GMM-LDA. (a)** Cluster 1: (bin 2~158), **(b)** Cluster 2: (bin 1~161), **(c)** Cluster 3: (bin 9~141), **(d)** Cluster 4: (bin 1~165), **(e)** Cluster 5: (bin 9~154).

The shape of these curves reveals the coherence of each cluster. For a quantitative comparison, we analyzed the angular span that encompasses 95% of the total word frequency for each topic. A narrower angular span signifies that the cluster is defined by a more concentrated set of "words", indicating a more coherent and distinct primitive. Conversely, a wider span suggests a more diffuse or ambiguous cluster. This analysis confirms that VC-LDA produces more coherent and well-defined primitives, providing a clear structural explanation for its superior performance.

In **Fig 20**, the included angle for each primitive cluster ranges from bin 1–63 to bin 15–134. In contrast, **Fig 21** shows that the primitive clusters obtained by GMM-LDA cover nearly all data bins. Clearly, VC-LDA is more effective in accurately identifying the similarities in the same primitive differences between different driving behavior primitives.

In summary, it can be concluded that VC-LDA effectively preserves the coupling relationships between variables and provides a better analysis of driving behavior primitives based on driving data. The superiority of VC-LDA is thus confirmed.

## 6. Conclusion

### 6.1. The work in this research

Extracting meaningful, semantically rich primitives from complex, multi-dimensional driving data remains a significant challenge. To address this, we propose a novel framework that extracts behavior primitives in two key steps: a behavior segmentation stage that incorporates the driver's subjective expectations on vehicle performance, followed by a primitive clustering stage that leverages inter-feature coupling relationships.

For behavior segmentation, we introduce a multi-type feature space that combines two distinct sets of variables. The basic feature set captures objective driving maneuvers and vehicle motion states, while the constructed feature set uniquely quantifies the driver's subjective expectations on vehicle driving performance. This holistic representation allows for a more accurate and comprehensive segmentation of continuous driving data into candidate primitives.

For primitive clustering, we developed the Variable Coupling-based Latent Dirichlet Allocation (VC-LDA) model. The core innovation of VC-LDA lies in its data discretization process. By analyzing the coupling relationships among the multi-type features, it constructs a coherent "vocabulary" of driving states, which is then clustered using the LDA framework. This method effectively prevents the creation of meaningless or overly diffuse data bins, leading to significantly more coherent and distinct primitive clusters.

In essence, our contributions are twofold. The multi-type feature space provides a more holistic representation of driving, while VC-LDA ensures that the resulting primitives are semantically meaningful. Together, this framework extracts driving behavior primitives with rich, multi-dimensional semantics, offering deeper insights into the causal relationships behind driving actions.

### 6.2. The future research

**6.2.1. Dataset expansion and diversification.** Future research will address the limitations of the current dataset, which was collected under controlled conditions, by focusing on two primary directions to enhance the variability and richness of driving scenarios. Firstly, a wider range of adverse weather conditions will be incorporated. This involves systematically studying the impact of rain, snow, and fog on driver behavior to improve the model's robustness under different meteorological conditions. Secondly, design more challenging and hazardous scenarios will be designed. This initiative will be twofold:

(1) For static road environments, test routes featuring special road sections will be designed, such as sharp curves, steep slopes, tunnels, and roads adjacent to water or cliffs, to evaluate driver behavior in extreme road conditions.

(2) For dynamic traffic environments, inspiration from advanced methodologies in autonomous driving scenario generation is drawn [38]. This includes designing interactive (game-theoretic) scenarios and edge cases to capture driver responses during unpredictable, high-risk interactions. The goal is to enrich the dataset to cover a broader spectrum of real-world driving behaviors.

**6.2.2. The research based on behavior primitives.** The concept of driving behavior primitives, defined as the smallest behavior segments with distinct physical meaning, opens up numerous avenues for future research. By decomposing long-term, continuous, multi-dimensional driving data into semantically rich primitives, a delicate analysis of driving dynamics becomes possible. The extracted primitives can serve as a foundational element for various advanced studies.

(1) *Driver-centric behavior analysis and coaching.* The extracted primitives can be leveraged to detect abnormal driving behaviors. By identifying and quantifying these abnormal primitives, it becomes possible to provide drivers with personalized feedback, including statistical insights on the frequency, timing, and location of risky events, thereby facilitating safer driving habits [39,40]. Furthermore, an analysis of primitive transition characteristics (e.g., transition probabilities and tendencies) can be used to model time-varying driving patterns and construct individualized driver profiles [41].

(2) *Interactive behavior analysis with surrounding vehicles.* When the data stream includes information from surrounding vehicles, the methodology can be extended to extract primitives with V2V interaction semantics. This enables a deeper analysis of the host vehicle's decision-making logic in complex traffic scenarios, an assessment of interaction patterns, and the prediction of potential decision-making failure points [42].

(3) *Scenario generation for ADAS/AV testing.* The interactive behavior primitives can be systematically concatenated to generate realistic and challenging test scenarios for ADAS and AVs. This provides a novel, data-driven methodology for creating a comprehensive library of test cases, particularly for edge cases that are difficult to replicate in simulated or physical testing environments.

The primitive-based approach provides a powerful analytical lens for both driver assistance systems and traffic safety research. These potential research directions collectively highlight the broad applicability and significant value of this framework.

## Supporting information

**S1 File. Driving behavior dataset for straight driving.**
(ZIP)

**S2 File. Driving behavior dataset for turning.** The dataset utilized in this study is organized into two main folders, categorizing driving scenarios into straight driving and turning maneuvers. These folders contain the raw, unprocessed behavioral data. For a detailed description of the variables and format within the.xlsx files, please refer to the accompanying readme.txt file.
(ZIP)

## Author contributions

**Conceptualization:** Xiaotong Cui, Xuelian Zheng, Xiansheng Li.

**Data curation:** Yuanyuan Ren, Xuelian Zheng.

**Funding acquisition:** Yuanyuan Ren, Jianfeng Xi.

**Investigation:** Xuelian Zheng.

**Methodology:** Yuanyuan Ren, Xiaotong Cui, Xuelian Zheng.

**Resources:** Yuanyuan Ren, Xuelian Zheng, Jianfeng Xi.

**Software:** Xiaotong Cui.

**Supervision:** Xiansheng Li.

**Validation:** Yuanyuan Ren, Xiaotong Cui, Xuelian Zheng.

**Visualization:** Xiaotong Cui.

**Writing – original draft:** Xiaotong Cui.

**Writing – review & editing:** Xuelian Zheng.

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
