## [Decision Letter · Decision Letter 0]

26 Sep 2025

Dear Dr. Zheng,

Thank you for submitting your manuscript to PLOS ONE. After careful consideration, we feel that it has merit but does not fully meet PLOS ONE’s publication criteria as it currently stands. Therefore, we invite you to submit a revised version of the manuscript that addresses the points raised during the review process.

We look forward to receiving your revised manuscript.

Kind regards,

Zhihong (Arry) Yao, Ph.D.

Academic Editor

PLOS ONE

Journal Requirements:

3. Please note that PLOS One has specific guidelines on code sharing for submissions in which author-generated code underpins the findings in the manuscript. In these cases, we expect all author-generated code to be made available without restrictions upon publication of the work. Please review our guidelines at https://journals.plos.org/plosone/s/materials-and-software-sharing#loc-sharing-code and ensure that your code is shared in a way that follows best practice and facilitates reproducibility and reuse.

4. Please note that funding information should not appear in any section or other areas of your manuscript. We will only publish funding information present in the Funding Statement section of the online submission form. Please remove any funding-related text from the manuscript.

“This research was funded by the National Key R&D Program of China (2023YFC3009600)”

6. In the online submission form you indicate that your data is not available for proprietary reasons and have provided a contact point for accessing this data. Please note that your current contact point is a co-author on this manuscript. According to our Data Policy, the contact point must not be an author on the manuscript and must be an institutional contact, ideally not an individual. Please revise your data statement to a non-author institutional point of contact, such as a data access or ethics committee, and send this to us via return email. Please also include contact information for the third party organization, and please include the full citation of where the data can be found.

Reviewers' comments:

Reviewer's Responses to Questions

**Comments to the Author**

1. Is the manuscript technically sound, and do the data support the conclusions?

Reviewer #1: Yes

Reviewer #2: Yes

2. Has the statistical analysis been performed appropriately and rigorously?

Reviewer #1: Yes

Reviewer #2: Yes

3. Have the authors made all data underlying the findings in their manuscript fully available?

Reviewer #1: Yes

Reviewer #2: Yes

4. Is the manuscript presented in an intelligible fashion and written in standard English?

Reviewer #1: Yes

Reviewer #2: Yes

Reviewer #1: 1. Although the abstract mentions improvements over the traditional LDA through the proposed VC-LDA approach, it lacks clarity on the evaluation metrics used and the nature of the dataset involved.

2. While the research gap regarding driver expectation and vehicle dynamics is well identified, the study fails to properly address the underlying cognitive or theoretical models that justify these constructs.

3. The use of the BMASS model for behavioral segmentation is a reasonable choice compared to traditional methods like change-point detection or entropy-based segmentation. However, no justification or sensitivity analysis is provided for this selection.

4. The paper lacks a thorough discussion of its limitations and fails to provide concrete and actionable recommendations for future work.

5. Although the result analysis is numerically sound, it remains shallow in behavioral interpretation. The influence of each behavioral cluster on driver behavior and its application to decision-making systems should be further discussed.

6. The behavioral clusters extracted using VC-LDA are labeled but not semantically interpreted. Their real-world behavioral significance is not addressed.

7. The modeling of driver expectations is unclear — it is not specified whether these expectations are based on geographic context, driver experience, or environmental conditions.

8. The potential applications of this research in AV and ADAS development, driver monitoring, driving education, and intelligent fleet planning are valuable. However, the paper lacks a direct and explicit discussion linking its findings to real-world industrial deployment.

9. The sample size is limited to a small number of drivers, and the driving scenarios lack sufficient diversity.

10. The model does not account for exogenous variables such as road conditions, weather, or traffic density, which significantly affect driver behavior.

11. The cluster interpretation remains general, lacking causal or behavioral depth. For example, what justifies the classification of a cluster as high-risk or defensive driving?

12. No sensitivity analysis is conducted on key model parameters (e.g., number of clusters, window size, discretization rate), limiting understanding of model robustness.

13. The extracted behavioral primitives are labeled but not functionally or semantically analyzed (e.g., what decision does a given cluster correspond to? Does it indicate responsiveness or aggressiveness?).

14. The methodology of the VC-LDA model lacks formal mathematical description, including assumptions, inference procedure, and coupling formulation.

15. No mention is made of participant consent or ethical approval, which is essential for studies involving human subjects — particularly for journals that adhere to strict ethical review standards.

Reviewer #2: This paper addresses the problem of extracting driving behavior primitives by proposing a segmentation and clustering framework that combines a multi-type variable space with an improved LDA-based method. The topic is relevant and valuable, especially for applications in autonomous driving and advanced driver assistance systems. However, several aspects require further clarification and improvement to strengthen the scientific soundness and persuasiveness of the work.

1. The paper employs simulated driving data (RADS 8 DOF), which is reasonable. However, the sample size is small (only 16 drivers), and it is unclear whether the dataset covers diverse driving scenarios (e.g., urban, highway). The authors are encouraged to provide more details about data diversity or to discuss the generalizability of the approach.

2. Some terms are used inconsistently, such as “multi-type variable space” and “multivariate space,” which appear interchangeably. It is recommended to unify terminology throughout the paper to avoid confusion.

3. Lines 234–248 describe the process of time window selection, but the justification for ultimately adopting a 1-second window is not clearly or convincingly explained. The reasoning appears to involve logical gaps and inconsistent standards. This section would benefit from additional theoretical or experimental support.

4. The references are somewhat outdated, with limited coverage of the last five years. It is suggested to add more recent studies, especially those focusing on driving behavior analysis, sequence segmentation, and clustering methods, to highlight the paper’s connection with the state of the art.

**Do you want your identity to be public for this peer review?** For information about this choice, including consent withdrawal, please see our Privacy Policy

Reviewer #1: No

Reviewer #2: No

---

## [Author Response · Author response to Decision Letter 1]

26 Nov 2025

We have prepared a point-by-point response to all the reviewers’ comments and have uploaded it as a separate file in the “Attach Files“ section.

---

## [Decision Letter · Decision Letter 1]

16 Dec 2025

Extraction of driving behavior primitives considering driver expectation and vehicle dynamics

PONE-D-25-29818R1

Dear Dr. Zheng,

We’re pleased to inform you that your manuscript has been judged scientifically suitable for publication and will be formally accepted for publication once it meets all outstanding technical requirements.

Kind regards,

Zhihong (Arry) Yao, Ph.D.

Academic Editor

PLOS One

Additional Editor Comments (optional):

Reviewers' comments:

Reviewer's Responses to Questions

**Comments to the Author**

Reviewer #1: All comments have been addressed

Reviewer #2: All comments have been addressed

2. Is the manuscript technically sound, and do the data support the conclusions?

Reviewer #1: Yes

Reviewer #2: Yes

3. Has the statistical analysis been performed appropriately and rigorously?

Reviewer #1: Yes

Reviewer #2: Yes

4. Have the authors made all data underlying the findings in their manuscript fully available?

Reviewer #1: Yes

Reviewer #2: Yes

5. Is the manuscript presented in an intelligible fashion and written in standard English?

Reviewer #1: Yes

Reviewer #2: Yes

Reviewer #1: Dear authors

Your revisions addressing the semantic definitions, contextual interpretation, and methodological clarity have substantially improved the manuscript, and I now recommend it for publication

Reviewer #2: This version of the paper makes a good revision of all the comments. I recommend this article to be published on PLOS One.

**Do you want your identity to be public for this peer review?** For information about this choice, including consent withdrawal, please see our Privacy Policy

Reviewer #1: No

Reviewer #2: No

---

## [Editor Report · Acceptance letter]

PONE-D-25-29818R1

PLOS One

Dear Dr. Zheng,

I'm pleased to inform you that your manuscript has been deemed suitable for publication in PLOS One. Congratulations! Your manuscript is now being handed over to our production team.

Kind regards,

on behalf of

Dr. Zhihong (Arry) Yao

Academic Editor

PLOS One